



# Diurnal mesospheric tidal winds observed simultaneously by meteor radar in Costa Rica (10°N, 86°W) and Cariri (7°S, 37°W)

Ricardo A. Buriti[1], Wayne Hocking[2], Paulo P. Batista[3], Igo Paulino[1], Ana R. Paulino[4], Marcial Garbanzo-Salas[5], Barclay Clemesha[3] (in memoriam), Amauri F. Medeiros[1]

[1]Unidade Acadêmica de Física, Universidade Federal de Campina Grande, C. Grande, 58429-900, Brazil
[2]Department of Physics and Astronomy, University of Western Ontario, London, N6A3K7,Canada
[3]Instituto Nacional de Pesquisas Espaciais, S. J. dos Campos, 12227-010, Brazil
[4]Departamento de Física, Universidade Estadual da Paraíba, C. Grande, 58429-500, Brasil
[5]Departament of Atmospheric, Oceanic and Planetary Physics, Universidad of Costa Rica, San Jose, 11501-2060, Costa Rica

*Correspondence to*: Ricardo A. Buriti (rburiti@df.ufcg.edu.br)

**Abstract.** This paper presents a study of diurnal tidal winds observed simultaneously by two meteor radars sited either side of the equator in the equatorial region. The radars are located in Santa Cruz (10.3°N, 85.6° W), Costa Rica (hereafter CR) and in São João do Cariri (7.4°S, 36.5° W), Brazil (hereafter CA). The distance between the sites is 5800km. Harmonic

analysis was used to obtain amplitudes and phases (hour of peak amplitude) for diurnal, semidiurnal and terdiurnal tides between 82 and 98 km altitude, but in this paper we concentrate on the diurnal component. The period of observation was from April 2005 to January 2006. The results were compared to the GSWM00 model. In general, seasonal agreement between observation and model diurnal tides was qualitatively satisfactory for both zonal and meridional amplitudes. However, magnitudes of zonal and meridional amplitudes from November to January for CR were quite different to the

predictions of the GSWM00. Peak zonal amplitudes (~25 m/s) at CR were observed in September and December between 90 and 94km. In regard to phases, agreement between meridional tidal phases at the two sites was excellent, and a vertical wavelength of 25 km for the diurnal tide was observed practically every month, although at times determination of the vertical wavelength was difficult due to non-linear phase variations with height. In regard to the diurnal zonal amplitude, there are notable differences between the two sites. This is probably because while the sites are somewhat complementary,

the local climate at the two sites are quite different , with CA being drier, so that the effect of heat latent release could vary between the sites, resulting in different responses at high altitudes. The possibility of significant non-migrating tides in Costa Rica is noted.

**Keywords:** MLT dynamics, meteor wind, diurnal tide.




## 1 Introduction

Atmospheric tides are driven principally by solar heating which results in significant day-night differential heating, and are dynamically very dominant at mesospheric heights. Thermal excitation due to absorption of solar radiation by water vapor (at infrared wavelengths) and ozone (at ultraviolet wavelengths), coupled with latent heat release at low altitudes, results in expansion and contraction of atmospheric pressure/density fields, creating modes of oscillation with very well-defined characteristics. Such oscillations are particularly easy to observe in the lower thermosphere through their impact on wind fields, pressure, temperature, airglow and other diagnostics. Because of this, tides are very important to the ionosphere-thermosphere system, and linear and non-linear interactions between solar atmospheric tides, gravity waves and planetary waves have been studied in order to better describe the dynamics of the atmosphere from low to high altitudes (e.g., Garcia and Solomon, 1985; Teitelbaum et al., 1989; Meyer, 1999, Thayaparan e al., 1995). Although the classical theory of tides is moderately well-established, many issues about interaction, excitation and temporal variability require further understanding. The presence of tides in wind fields observed by various methods (including meteor radar), has shown good agreement with the GSWM Global Scale Waves Model in some cases (Chang, et al., 2012, Hagan, et al., 2002, 2003). Previous studies about tides in the equatorial region have shown that, in the altitude-range between 82 km and 98 km, the diurnal (24-hr period) amplitude is more significant than semidiurnal mode for both zonal and meridional components (Buriti, et al., 2008; Davis et al., 2013). Tides also have a dependence on altitude and season. That behavior is in accordance with tidal theory for the propagation of the (1,1)Hough mode (Chapman and Lindzen, 1970; Forbes, 1982). Frequently, the meridional diurnal mode presents a better-defined behavior as a function of altitude and season, which makes the calculation of the meridional vertical wavelength more accurate relative to the zonal component. The semidiurnal mode (period of 12 hours) is generally weaker than the diurnal mode in the equatorial regions. The terdiurnal and quadiurnal tides are also present but with even smaller amplitudes, but nonetheless do play some role in mesospheric atmospheric dynamics (e.g. Guharay et al., 2018).

This paper concentrates on diurnal tides observed simultaneously with meteor radars installed in Costa Rica and São João do Cariri, Brazil, with our focus being on the period from April 2005 to January 2006 (inclusive). Both radars, separated by 5800 km, are very similar, and they are located in opposite hemispheres but very close to the equator. Their latitudes are almost complementary. This is the first time that meteor winds from Costa Rica have been simultaneously compared to Cariri observations and the GSWM model. The paper first presents a brief overview of the background wind at both sites, and then proceeds to a comparison between tidal characteristics. Amplitudes are discussed first, followed by phases. A discussion then follows.

Interesting results include a peak in amplitude observed in the diurnal zonal amplitude at the Costa Rican site in December which is not predicted by the model, and a clear anti-phase between Costa Rica and Cariri in regard to the diurnal meridional component, which is predicted by the model.





## 2 Instruments and Observation

The meteor radars used are called SKiYMet radars. These are All-Sky Interferometric meteor radars which consist of a transmitter antenna in the form of a 3-element Yagi, and a set of 5 receiver antenna comprising 2-element Yagis. The radars

are installed in different locations, namely in São João do Cariri, PB, Brazil (7.4°S, 36.5°W) and Santa Cruz, Costa Rica (10.3°N, 85.6°W). The distance between the sites is about 5800 km, and they are at similar latitudes either side of the equator (10°N and 7°S). The first uses a frequency of 35.24 MHz and the second one operates at 35.65 MHz. The radars run 24 hours per day without interruption, and provide meridional and zonal wind data at altitudes between 80 and 100 km. Weather conditions do not interfere with observations. Basically, the wind is measured when an ionized meteor trail, formed when a

meteoroid collides with the atmosphere, reflects the radio-wave emitted by the transmitter antenna. The echo is detected by 5 receiver antennas. The phase-shift between each pair of antennas gives information about the direction in which the meteor trail was observed, the time delays of the transmitted pulses give the range to the target, and the Doppler shift of the received signal gives the radial velocity. Typically several thousand meteor trails are detected per day. This combination of data allows generation of a wind-field as a function of height and time (Hocking et al., 2001). The temperature of the mesosphere

at the height of peak meteor detection (~90 -92 km) can also can be determined by meteor radar (Hocking, 1999), but we will concentrate on the wind field. In our case, we determine information of winds every 2 hours centered at altitudes of 82, 85, 88, 91, 94 and 98 km.

In the present work, we will use Costa Rica data (hereafter CR) corresponding to the period from 14th April 2005 to 29th

January 2006, with a gap of data from 17th November to 13th December. Data from Cariri (hereafter CA) for the same period will be presented for comparison. A study of one year of background mean winds, as well as diurnal and semidiurnal tides observed in both the zonal and meridional components above CA during 2004-2005 has previously been reported by Buriti et al., (2008). We also include results of the Global Scale Waves Model (GSWM00).

### 2.1 Background winds
In order to set up the background conditions for the tides, we will here present wind variations on scales of months, and try to determine annual and semi-annual variability. Fig. 1 shows the monthly averages of zonal (left) and meridional (right) winds in CA and CR. Data for February and March are missing for Costa Rica, so some caution is required in interpreting annual variations. Comparing monthly mean winds at the two sites, some interesting results are evident. In general, both sites

seem to present a clear semiannual behavior, particularly in regard to the zonal wind. At heights of 82-91 km, the maximum eastward mean wind at CA is observed in June, while the maximum at CR is present in December. This is almost a 6 month delay, as might be expected due to the fact that the radars are in different hemispheres. The meridional winds are quite different at the two sites, although strong southward flows above Cariri in June-July and strong northward flows in December over Costa Rica are evident.




A long-term yearly harmonic analysis was carried out in regard to these data, involving annual and semiannual components (Fig. 2). Considering all data between April 2005 and January 2006 inclusive, it was shown that the semiannual zonal amplitude decreased between 82 and 94 km in both sites by about 2 m/s/km. At 82 km the semiannual zonal amplitudes over CA and CR were 28.1 m/s and 23.4 m/s, respectively. Those values decreased to about 4 m/s at 94 km height. The

meridional component, on the other hand, was almost 5 times weaker. The meridional amplitude above CR decreased from 5.8 m/s at 82 km to about 1 m/s at 98 km. Semiannual zonal phase (phase indicating the maximum value) over both sites were also very similar and constant between 82 and 91 km. The time of maximum is close to June 9th (160 doy). On the other hand, while recognizing that some care in interpreting annual analysis is needed, the annual amplitude at both sites decreases by about 1.4 m/s/km, but CA amplitudes were always larger than for CR. Concerning the annual phase, the CA

time of maximum is, on average, close to July 19th (200 doy) while for CR it is on about September 17th (260 doy).

It is clear that, during the period of observation, the meridional wind over CR was almost always northward, while in CA, it was southward between May and September.

While the semiannual zonal wind varies only modestly in phase between 82 and 98 km, discussion of the meridional wind needs some care. The meridional semiannual amplitude is present with very low values when compared to the zonal wind. For example, CR presents amplitude between 6 m/s at 82 km altitude, and 1 m/s at 98 km. The amplitude for CA, on the other hand, decreases from 2 m/s at 82 km to 1 m/s at 91 km but increase to 5.4 m/s at 98 km height. The annual amplitude in CA is around 10 m/s between 85 and 98 km while CR is 3.5 m/s. A curious fact is the annual amplitude of both sites

present the same value at 82 km, 8.9 m/s. Concerning the annual phase, CA and CR present similar results. On average, the phase is close to 6 doy (January 6th) between 82 and 98 km.

### 3 Diurnal tide

We now turn to tidal analyses. The analysis of Costa Rica and Cariri winds in order to determine information about the diurnal tides was similar to the procedure described in Hocking (2001) and Buriti et al. (2008). First of all, a superposed

epoch averaging of winds at two-hour steps was made, producing monthly means at 0100, 0300, ... 2300 hours local time. After that, a standard least-squares fitting technique was used to obtain amplitude, phase and mean values for each month. It is known that the diurnal oscillation of the meridional wind in regions close to the equator present good regularity in amplitude and phase according to altitude, and our results confirmed this. Because of this, a precise vertical wavelength is easier to calculate for the meridional wind than for zonal wind. A very interesting observation can be made regarding the

diurnal phase of the meridional wind at the two sites. They are completely out of phase. In other words, if the wind has maximum intensity to the south in CA, then at the same local time in CR, the meridional wind has maximum intensity to the north. This is predicted by the GSWM.



In Figs. 3, 4, 5 and 6, information about amplitudes and phases of the GSWM and the radars are presented for 6 different

altitudes. The altitudes used for the GSWM do not coincident exactly with the specific altitudes of the radar, but nonetheless the comparisons between radar data and the GSWM are still easy to make. We now turn to more detailed discussions, beginning with the zonal diunal tide.

### 3.1 Zonal diurnal amplitude

A general view of the observational diurnal tidal amplitudes at CR and CA, as well as the GWSM00 at both sites, can be

seen in Fig. 3. In CR and CA the mean amplitudes were close to 10 ± 5.7 m/s, but there is a clear difference between them. While CR values were above the average for November-January at all altitudes, CA values were largely below the average for altitudes between 82 and 91 km height for the whole period of observation. Also, amplitudes at CR were small between 82 and 98km for May-July, as predicted by the model. CA presented similar results in October-January, but with a 6 month delay. Comparing to the model, CA is closer to the model. The small amplitude predicted by the GSWM00 in December-

January is not observed over CR. The presence of large amplitudes in September seems to be common between the sites. In CR the amplitude increased to values ~24 m/s at altitudes ~94 km in September and December. On the other hand, CA presented values above 18 m/s between 91 and 98 km (32 m/s) in September. On average, considering the dependence of amplitude with altitude, the amplitude in CR increased from 82 (7.8 m/s) to 91 km (15 m/s), then decreased until 98 km (6.6 m/s). CA presented a minimum at 85 km altitude (5.8 m/s), and increased almost linearly to about 15.6 m/s at 98 km.

### 3.2 Meridional diurnal amplitude

The meridional diurnal tide predicted by the GSWM at both sites is very similar. Observationally, CR presents larger amplitudes compared to CA. But both sites show, according to the model, amplitudes above 20m/s in April and October at altitudes between 82 and 102 km, and minimum amplitudes in July and December-January. Comparing CR and CA, it is clear that they are similar to each other in this regard. The amplitudes increase in July after presenting a minimum in May-

June. On average, considering data from April to January in the range of 82 to 98 km of altitude, diurnal amplitudes at CR and CA were 23.3 ± 11.4 m/s and 22.6 ± 9.2 m/s, respectively. In CR, November and December presented the maximum values for altitudes above 85 km. It is clear the maximum values occurred close to 91-94 km. This is different to the GSWM, where the maximum seems to be above 100 km height in CR and CA. On the other hand, CA presented maximum amplitude close to 40 m/s in April and October at altitudes of 88-91 km. Meridional amplitude presented values close to the double the

values of both the predicted and observed zonal amplitude.





### 3.3 Zonal diurnal phase

The zonal phase (in Local Time) presented interesting results. These included a clear uniform phase difference in altitude at
CR, except for January 2006; a large variation of phase at CA for altitudes of 82-91 km from July to January; May-July and
November presented only small variations of phase suggesting perhaaps a dominant evanescent mode or non-migrating tide.
Fig. 5 shows the observed phases at different altitudes in CR and CA as a function of month of the year. The phase in CR, in
contrast to CA, shows a clear linear dependence on altitude in most months, which makes it possible to determine the
wavelengths of the tidal propagation assuming a quasi-monochromatic wave. A decrease of the phase between May and
January is generally evident. Also, an upward propagation is clear, especially in CR. The vertical wavelength was obtained
considering the altitude as an independent variable, but some additional criteria were considered in order to extract reliable
vertical wavelength. In particular, a linear regression of at least 4 altitudes in sequence was required, and fit was only
accepted if the R-squared value was above 0.9. The results for CR and CA, on average, were $25.4 \pm 4.0$ km and $22.7 \pm 7.3$
km respectively. Because the zonal diurnal phase at CA showed evidence of modal superposition between the altitudes, only
4 months were available to determine the vertical wavelength using the above criteria. According to the GSWM, the vertical
wavelength in CR and CA should be about 29.8 km. It is almost 20% higher than that one observed in CR. We discarded
very long vertical wavelengths in our analysis not because they could be indicative of evanescent structure, but simply
because the criteria discussed above were not satisfied. Nevertheless, it seems that the GSWM does overestimate the vertical
wavelengths.


The zonal diurnal phase, according to the GSWM, has little appreciable difference in values and behavior between CR and
CA. Typically, any differences are less than about 0.2 hour. Hence in Fig. 5 we only plot the phase at CR, since CA is so
similar. With regard to observational comparisons, there is some similarity of phase between CA and the GSWM at altitudes
in the range of 91 and 98 km from April to September, possibly because the diurnal amplitude is large there. Agreement is
not so good below 91 km. Comparing observed and model results, we can conclude that in general the GSWM does not
reproduce well, in terms of values, zonal diurnal phase to low latitudes.

### 3.4 Meridional diurnal phase

The meridional phase (in Local Time) presents a behavior quite different to that of the zonal component. Fig. 6 presents
meridional phase observed by radar and by GSWM at CR and CA. It is clear that a downward phase propagation is evident
at both sites, and a small decrease of phase from May-June to January occurs. The regularity of phase with altitude permitted
us to estimate the vertical wavelength, using the criteria presented previously, for all months with data. The results were
vertical wavelengths of $25.1 \pm 5.3$ m/s and $25.6 \pm 4.6$ m/s at CR and CA, respectively. According to GSWM, the vertical
wavelengths at CR and CA should be 26.9 km and 27.5 km. An interesting thing that was observed was the difference of
phase, in local time, between CR and CA. At the same altitude and month, the difference in the time of maximum at CA



compared to CR was 13.3 ± 2.3 hour, on average, if we consider that CR is ahead of CA.  If we concentrate on altitudes

between 85 and 94 km, then that value, on average, goes to 12.2 ± 1.6 hour from May to October. So the phases are close to

12 hours different. This means that when the meridional wind is maximum blowing to the south at CA, it is almost

maximum to the north in CR. Considering phases calculated by GSWM between 82 and 98 km, the difference between CA

and CA should be, on average, 12.0 ± 1.6 hours.

**4 Discussion**

Our work has focused on diurnal tides. We also have information pertaining to semidiurnal and terdiurnal tides, but these

results were not included in this work. The observational results were compared to the Global-Scale Wave Model (GSWM),

version 2000. All data about the diurnal component were downloaded from the High Altitude Observatory (HAO/UCAR)

where the home page was http://www.hao.ucar.edu/modeling/gswm/  (at present we must replace www by www2). Briefly,

this software is a 2-dimensional model that solves the linearized and extended Navier-Stokes equations for a particular

period and wavenumber $s$ as function of latitude (from 87°S to 87°N), altitude (from 0 to 124 km) and month (from January

to December). It incorporates fields of wind (zonal), pressure, temperature and other important physical parameters from

empirical models, such as MSISE90 (Hedin, 1991). Depending on the altitude range, information on wind comes from

different models and satellite observation. For example, between the stratosphere and the mesopause, winds are provided by

High Resolution Doppler Interferometer - HRDI - on board the UARS satellite. Details about GSWM can be obtained on

HAO's homepage and a vast number of papers, such as Hagan et al, 1997; 2002; 2003; Manson et al., 2002;Pancheva et al.,

2001). Information about tidal parameters determined by GSWM are presented at specific altitudes that are not exactly

coincident to the radar heights (e.g. see Fig. 5 and 6), but they are close.

According to the GSWM00, zonal diurnal amplitudes at low latitudes present maximum peaks close to 30°S and 30°N

between altitudes of 82.1 and 98.7 km. The meridional component presents a peak at ~ 20°. This structure is attributed

particularly to latent heat released in the Troposphere. Close to the equator, the amplitudes of the two orthogonal

components are at a minimum. At ~82.1 km that minimum is at 3°S in June, winter at South Hemisphere. On the other hand,

in December the smallest value is at 3°N. Similar behavior happens for the meridional amplitude.


Comparisons of zonal diurnal amplitudes between radar and model data for specific months and altitudes are presented in

some detail in Fig. 7. The results essentially summarize Fig. 4 but with fewer specific details.   There is some difficulty in

identifying a pattern of similarity between radar and model when we analyze Fig. 7. For example, in regard to the zonal

amplitude, August showed excellent agreement between the model and observations at CA at both  82 and 91 km, yet CR

showed a large difference at 82 km but agreed well at 91 km. In October, the model reproduces very well the observations at



CR and CA at 82 km, but agreement is much poorer at 91 km. In April, there is an excellent agreement to CR, but only at 91 km. In December, the observation in CR was very different from the model at both altitudes and both components.

The meridional amplitudes, like the zonal ones, presented good agreement only at a specific altitudes and months, e.g., April
and June, and August showed an excellent agreement to CA at 82 km and to CR and CA at 91 km height, respectively. If we look back at Fig. 3, which shows the zonal amplitudes, some coincidences exist between the model and observation. For example, there is an observed peak in amplitude in September at 91-94 km in CR; a similar peak is predicted by the model, but at a higher altitude of 98 km. On the other hand, in December, a peak of amplitude at 91 km was observed in CR but it is not predicted by the model.


Both model and observation present a minimum in June but with different values. CA observations seem to be closer to the model than does CR. An increase in amplitude with increasing height is quite common during most months of observation. The meridional component, shown in Fig. 4, also presents some discrepancies between model and observation. For example, the model predicts a strong increase of amplitude in October over CR, and a noticeable but weaker increase at the same time
in CA. Such an increase was seen not in October but rather two months later (in December) at CR, and not at all over CA. In June-July, the observation at both sites presented a minimum but with different behavior compared to the predictions by the model,  which presents a clear semiannual behavior.

In general, the GSWM00 predicts the meridional component more satisfactory than the zonal one. However, just as for the
zonal amplitude, the meridional component also showed large differences between model and observed results in most months. On occasion, a good concordance did happen, e.g., CA at 82 km in April and June and CR at 91 km in August and October agree well with the model. When comparisons are made between the sites, June and July present similar results for zonal and meridional amplitude, respectively. In August and October, the zonal wind at 82 km in CR and CA are similar in magnitude.


Davis et al., 2013, reported a study of diurnal amplitude of meteor wind observed at Ascension Island (8°S, 14°W) from 2002 to 2011. They show in Fig. 6 of their work a composite-year monthly wind of zonal and meridional diurnal amplitudes which present a good agreement to CA and an excellent agreement to the GSWM00 model. The similarity to the GSWM may arise because the site was an island, so local ground-level variability may be less.  In addition, the availability of 9 years
of data may have smoothed out irregularities that arise in any one year. In that work, they also have compared their observation with the Canadian Middle Atmosphere Model (eCMAM) and the Whole Atmosphere Community Climate Model (WACCM) (Fomichev et al., 2002; Du et al., 2007).





The differences between CA and CR may be due to local orography and climate. Geographic conditions at CR and CA are
quite dissimilar.  It would be reasonable to expect different behavior of the tides because of their different levels of forcing
due to water vapor and tropospheric latent heat release. Specifically, the climate is desert-like at CA but very tropical in
Costa Rica. Costa Rica is located in Central America. It is a country of width ~120 km from southwest to northeast,
surrounded by the Atlantic (east) and Pacific (west) Ocean. São João do Cariri, on the other hand, is a city in the country of
the Northeast of Brazil, having the Atlantic Ocean 190 km to east and 250 km to north.  The Pacific Ocean is 4800 km to the
west. CA is located in the driest region in Brazil. Some reports have proposed that latent heat release is important to
semidiurnal tides. Lindzen (1978) originally considered that latent heat release is not important to the diurnal tide. Since
then, however, many reports about the possibility of diurnal tides, including migrating and non-migrating, in the MLT being
affected by ground-level  sources in the tropical region have been  published (Hamilton, 1981; Hagan, 1996; Forbes et al.,
1997). Hagan et al., (1997) showed the importance of the seasonality of convective activity in the Troposphere to the diurnal
amplitude of the meridional wind at 21°N; it is strong in January and weak in July. This clear dependence is due to the
diurnal amplitude of the effective rainfall rate that varies with months. So, convective activity could explain the difference of
zonal component behavior between CR and CA. Ascension Island, which is located practically in the middle of the Atlantic
Ocean, has a desert climate with total precipitation only 200 mm per year. This is almost half that of CA, and ten times less
than CR. As we have presented above, CA (and Ascension) tends to be closer to the model predictions than CR. It is likely
that others modes of oscillation (including non-migrating tides), which are more sensitive to latent heat release, are present in
the Costa Rica winds.  Another important consideration is how close the source of latent heat release must be from the site of
observation. Probably the high precipitation in Costa Rica could contribute to diurnal tides over the meteor radar site
installed in Santa Cruz. We did not have sufficient information to estimate the contribution of non-migrating components to
the diurnal tides observed in this work, but we believe that there is a clear evidence that these components are likely to be
more dominant in CR than in CA.

## 5 Conclusions

Ten months of simultaneous observation of mesospheric winds by meteor radar installed in Costa Rica and Cariri have been
analyzed in order to compare tidal winds, with emphasis here on the amplitudes and phases of zonal and meridional diurnal
tides. A comparison of these observed parameters have been made to those predicted by GSWM00 model. Background
winds were also presented in the work. The monthly zonal winds in Cariri presented a semiannual oscillation similar to
Costa Rica, with values of amplitude close to 24 m/s at 82 km, decreasing to ~4 m/s at 94-98 km height. The meridional
winds, on the other hand, were small relative to the zonal ones, at least for altitudes above 85 km at both locations. The zonal
and meridional diurnal tidal parameters showed interesting results. CR presented a peak in diurnal zonal amplitude in
September at 94 km, in general agreement with the GSWM model. December also presented a peak at 91 km but, in contrast
to the September case, this was not predicted by GSWM. With regard to the meridional winds, observations at CR presented



a peak at 94 km height in December, while the model predicted a peak much earlier, in October. CE showed no strong activity at all in the meridional tides in the September to December time-frame. In a general way, diurnal meridional parameters measured over Cariri compared better to the GSWM. Vertical wavelengths measured at the site were often in broad agreement with the GSWM, being in the range 25-30 km, with the observational data showing slightly shorter vertical
wavelengths. Detailed comparisons (summarized in Fig. 7) showed periods of good agreement and periods of poor agreement with the GSWM. The case has been made that the very different climates at the two sites (with Cariri being desert-like and Costa Rica being very tropical) may be producing significant non-migrating tides, especially over Costa Rica. A longer-term study over many years may help clarify this possibility.

***Data availability****:* All meteor radar data can be requested from INPE and UWO. Contact Dr. Paulo Batista (paulo.batista@inpe.br) and Prof. Wayne Hocking (whocking@uwo.ca)

***Authors contributions:*** Ricardo Buriti is responsible for the operation of the radar in Cariri and has written the manuscript and made the analyses of data using software provided by Wayne Hocking. The same software is used  on the Costa Rica
meteor radar, which was also built and by W. Hocking using grants from NSERC in Canada.. Paulo Batista and B. Clemesha (in memoriam)  is responsible for the data and for the meteor radar of Cariri. I. Paulino, A. Paulino and A. Medeiros have contributed to the discussion of the manuscript. M. Garbanzo-Salas is a collaborator of W. Hocking and responsible for operation of the Costa Rica meteor radar.

***Competing interests:*** The authors declare they do not have any competing interests.

### Acknowledgements

This work is partially supported by National Council for Scientific and Technological Development (CNPq) under process 307247/2016-7. We also thank the Federal University of Paraíba for facilities to install and to operate the meteor radar.
Construction and support of the Costa Rica radar was supplied by the Natural Sciences and Engineering Research Council of Canada. The Land and buildings at the Costa Rica site were supplied by the University of Costa Rica.



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

        **in m/s.**





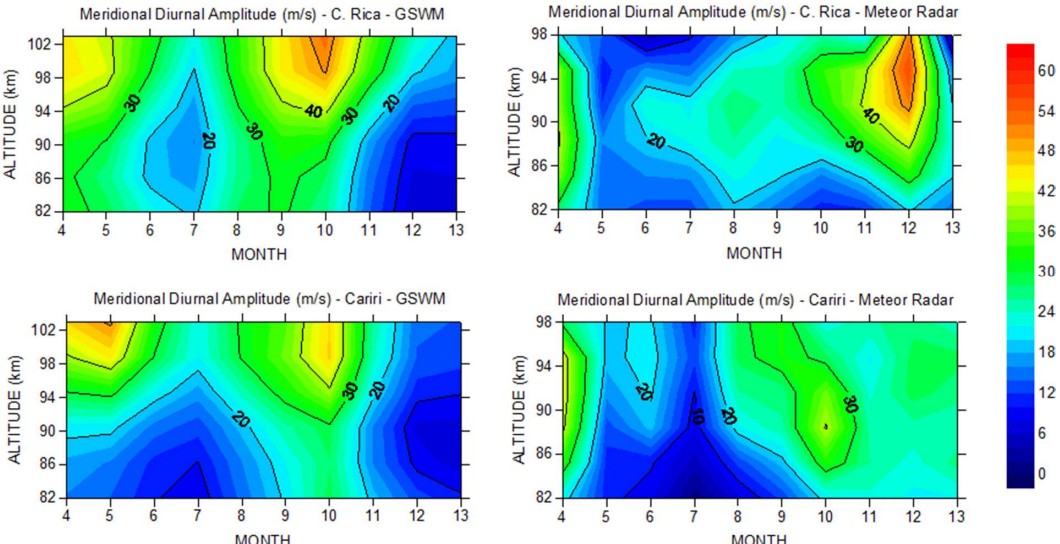

Figure. 4: Same as Fig. 3, but for meridional amplitude. The color scale is the double that of the Fig. 3.

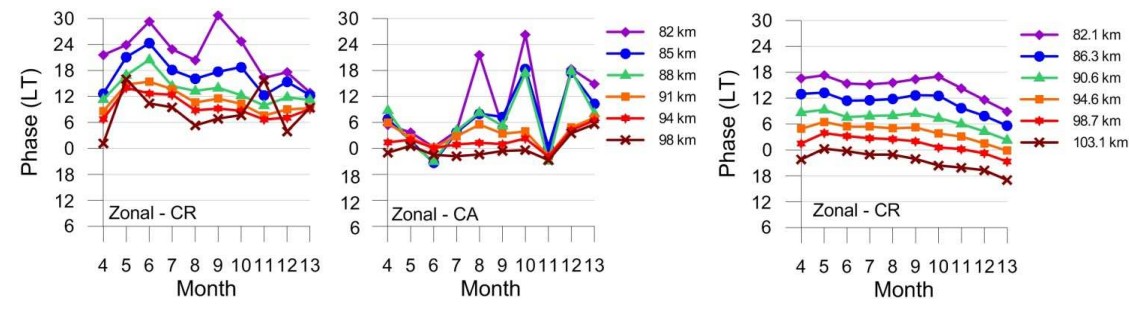

Figure. 5: Zonal diurnal phase in LT for CR (left), CA (middle) and GSWM (right) from April 2005 to January 2006. The height gates for the radar data and the GSWM data are not quite the same, but close enough for visual comparisons.




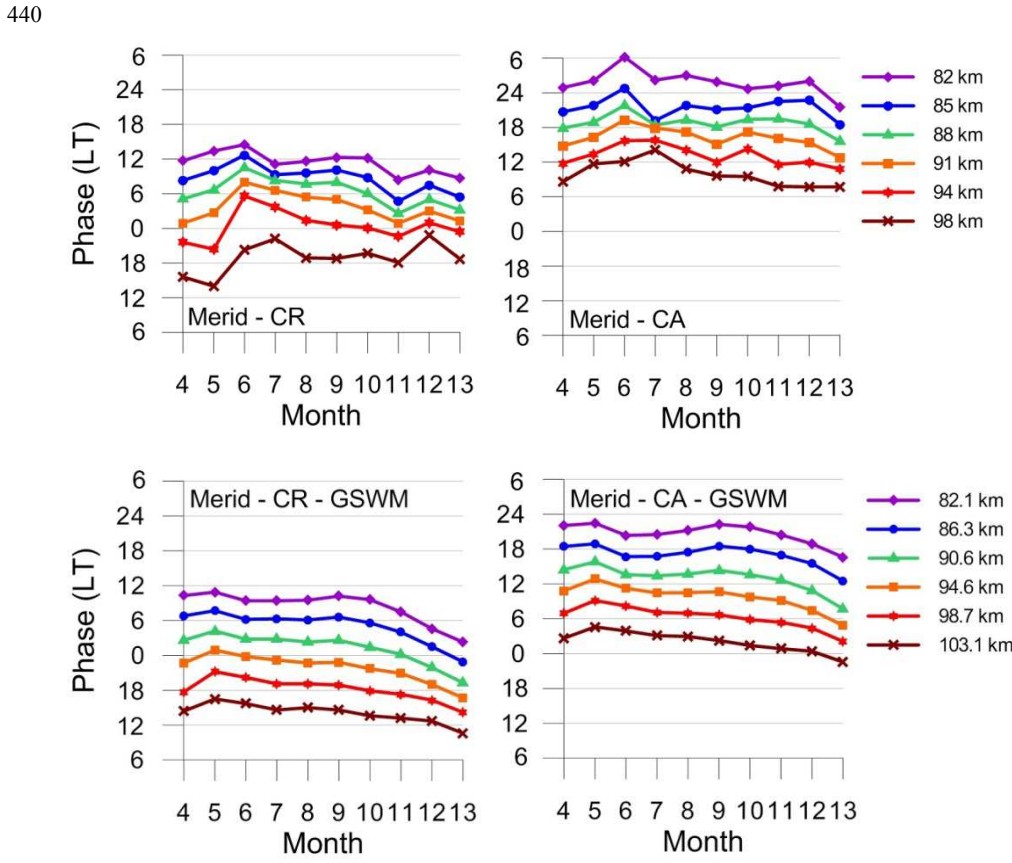

Figure. 6: Meridional diurnal phase (LT) observed at CR and CA (top panel) and diurnal phases calculated by GSWM for CR and
        CA. The height gates of the observed and calculated data are slightly different, but nonetheless the trends and similarities in
        observed data and GSWM data are clearly evident.





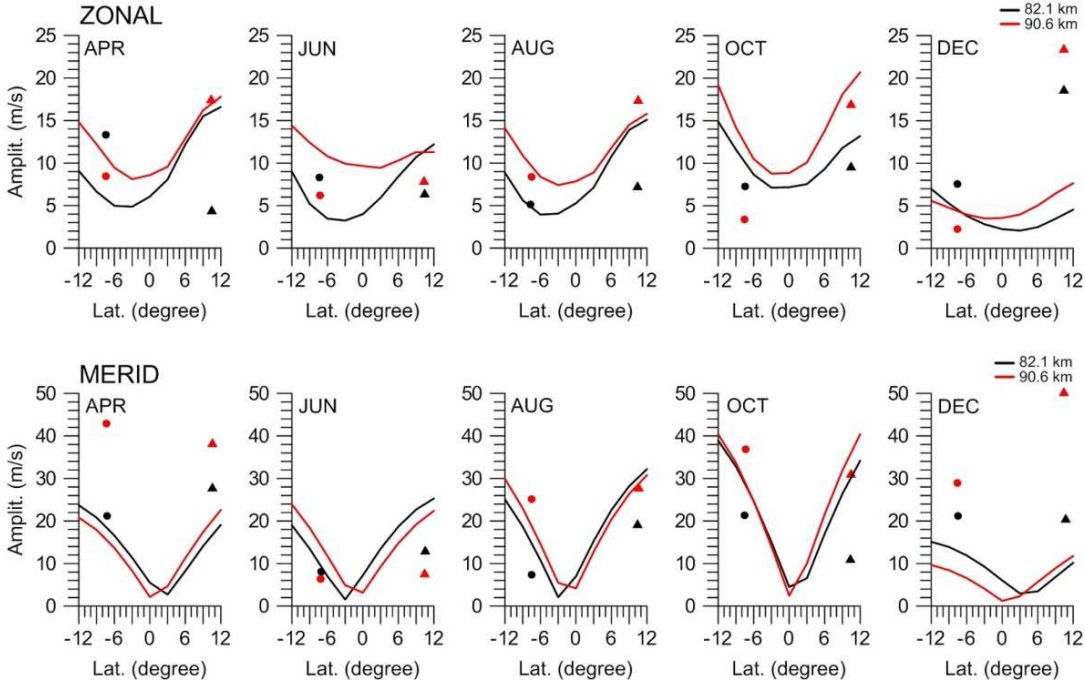


**Figure. 7: Diurnal amplitude calculated by GSWM between -12° and 12° at 82.1 km height (black curve) and at 90.6 km (red curve) to zonal (top) and meridional components. Full triangle and full circle represent observed amplitude to CR and CA at 82 km (black) and 91 km (red), respectively. Amplitude scale to meridional amplitude is 2x the zonal amplitude.**
