# Peer review of "Diurnal mesospheric tidal winds observed simultaneously by meteor radar in Costa Rica (10°N, 86°W) and Cariri (7°S, 37°W)"

_Annales Geophysicae, 2019_

## Referee Comment (RC1) · Anonymous Referee #1 · 24 Dec 2019

Review report of the manuscript entitled "Diurnal mesospheric tidal winds observed simultaneously by meteor radar in Costa Rica (10°N, 86°W) and Cariri (7°S, 37°W)" by R. A. Buriti et al. The paper presents comparative features of mean winds and tides in the MLT between two low latitude sites located in the opposite hemispheres using meteor radar winds. They further compared the observations with GSWM model. The topic of the present paper is interesting to the scientific community. However, the present manuscript contains several serious issues pertained to technical and scientific aspects. In view of presentation and language/vocabulary it can be considered as a draft and requires substantial modification to improve it to a communicatory level. Anyways, I am detailing my comments/suggestions below.

**Major points**

The theme of the paper is not clear as the title/abstract/conclusion points out the diurnal tide characteristics, but the results start with SAO and AO of the mean winds. Authors should decide the theme and organize the manuscript accordingly.

In the present comparative study the observational interval is too short (9.5 months) to present seasonal behaviour over a year which is another weak point of the paper. Derived AO features are questionable, especially at CR due to short data length. Since CA and GSWM database is longer, results can be shown for the missing months of the year.

Fig. 2: It seems that the authors carry out least square fit considering both SAO and AO simultaneously in the fitting function (as found in the caption of the figure). It is not clear how the authors decipher the individual SAO/AO amplitude/phase from the figure.

Fig. 7: Here GSWM data are shown over whole latitude range of ± 12°, which is not necessary. Also, authors limit to only two height bins. Instead of the present figure the authors can show four subplots using contours, estimating the difference between radar and GSWM amplitudes incorporating total MLT range of zonal/meridional for CR/CA. Deviation of amplitude can provide better clarity regarding the scientific point authors attempt to express.

Language needs significant improvement to bring coherence in the results/interpretations. It hinders spread of the essence of the work to the readers. Vocabulary, tense and preposition should be corrected. It will be a good idea to check with a native English speaker.

**Other points**

L. 21: "In regard to phases, agreement between meridional tidal phases at the two sites was excellent". The statement is incorrect. The meridional tide phases of two sites are almost opposite (~ 12 h difference).

Fig. 1: Assign marks, i.e., a, b, c, d

Fig. 2: Wind data should be shown along with the fit.

L. 96-116: The amplitudes of SAO and AO contain temporal variability. Authors' statement of specific amplitude at a particular altitude raises confusion as it does not make any sense. Same applies to phases.

L.25: correct "heat latent release" to "latent heat release"

L. 37: Replace "diagnostics" by "parameters"

L. 42: Correct "has shown…." to "showed…."

L. 44: Correct "have shown that…." to "showed that…."

L. 51: Delete "atmospheric" from the statement "mesospheric atmospheric dynamics…"

L. 86: Correct "variationson scalesof months…" to "variation on the time scale of month…"

L. 96: What do the authors mean by the term "A long-term yearly harmonic analysis"?

Amplitude of SAO and AO are found to be very small ~ 1-2 m/s. Information related to the uncertainty of the radar winds should be discussed in section 2.

Do the values shown after "±" represent standard deviations?

L.116: "On average, the phase is close to 6 doy (January 6th)". The meaning is not clear as the range in Fig. 2 is shown within 90-390 doy.

L. 136: correct "maximum intensity to the south…" to "maximum magnitude towards south…"

L. 140: correct "do not coincident…" to "do not coincide…"

L. 183: "upward propagation…". Please mention upward propagation of what.

L. 204: Correct "presented previously" to "mentioned ealier"

L. 206: Correct "thing that was…" to "feature…"

L. 210: Delete the statement "So the phases are close to 12 hours different."

Discussion: The first paragraph provides information related to the GSWM and it should be shifted to section 2 with some modifications.

L. 301: "Some reports have proposed that latent heat release is important to semidiurnal tides." Please provide reference.

L. 307: "So, convective activity could explain the difference of zonal component behavior between CR and CA". Since the tide is more prominent in the meridional wind such influence of convective activity should also be visible in that. Authors' claim of convective effect only on zonal component is not acceptable. Similar statement is also mentioned at the end of the abstract section.

The abbreviations CA and Cariri are used interchangeably throughout the manuscript. Please adhere to either CA or Cariri. Same applies to Costa Rica.

---

## Referee Comment (RC2) · Anonymous Referee #2 · 1 Feb 2020

The paper presents some new radar wind measurements in equatorial mesosphere and discussion on the diurnal tidal modulations in zonal and meridional winds. Although the description is relatively clean and easy to follow, my major concern is the comparisons with GSWM. The model version utilized in the paper is GSWM00 that does not include non-migrating tide. However, in equatorial region, the some non-migrating diurnal tides, such as DE3 and DE2, are equally important compared with diurnal tidal component, as the author mentions in the paper. Some of the major discrepancies in the paper are most likely due to this issue. There are more complete GSWM versions available, such as GSWM02, GSWM09 etc. that have non-migrating tidal components included. I strongly suggest the author to include the latest model

predictions in the next version of the paper, in addition to the current GSWM00 comparison, and expand the discussion based on the new comparison results. Minor comments: 1. line 24-26. I do not think the local weather can affect the tidal feature that much in mesosphere at the same location, since the tidal waves are global scale waves, propagating horizontally with very fast phase speed. The local observations in upper atmosphere reflect the tidal forcing several thousand km away, so the connection to the local tropospheric weather is not straight forward. 2. Line 43. I would add some more references on the tidal comparisons work between ground-based measurements and model. Here are a couple of them: Ward et al., 2010 and Yuan et al., 2006. 3. Line 111. "very low value" sounds strange, may be replaced with "less value". 4. Line 127. Maybe consider to add some brief discussion, from theoretical point of view, why there is such big difference between southern and northern hemisphere equatorial region. 5. Line 196-197, please consider to remove "We also have ... not included in this work" 6. Line 254, see my comment above. Unless there is some solid reference on this topic, I would hesitate to make such statement.

---

## Author Comment (AC1) · 31 Mar 2020

First of all, we appreciate a lot the comments and suggestions in order to improve the manuscript we are trying to publish. At this moment we want to say that this work is a good opportunity to compare simultaneous data of two meteor radars located at about opposite latitudes. We are not making a climatology of diurnal tides because, and we agree, the number of months of Costa Rica is not appropriate to this kind of analysis. But we believe that study, even limited by the number of months, should be present to the community. Unfortunately Costa Rica worked well for only 10 months in 2005-2006.

We request you to kindly permit us to extend the time to us to present a new version

of the manuscript in order to attend also the referee #2, that is, to include results of GSWM-09 in this work. So, a new analysis must be done and results can change and, naturally, a new text should be done to a new version of the manuscript. All of your comments and suggestion will be considered in the new manuscript.

Best regards
* * *

---

## Author Comment (AC2) · 31 Mar 2020

First of all, thank you for the comments and suggestions in order to improve ou manuscript. We believe the main point you have risen is to compare data of meteor radars to GSWM-09 simulation. We want to inform you the results of GSWM-09 is available to us at present. We are working on it in order to start the comparison very soon. Because of this, we kindly request more time to prepare a new version of the manuscript including GSWM-09 simulations.

Thank you very much

---

## Author Response (AR1)

Angeo-2019-134

**Response (*in italic*) to Referee #1.**

*First of all, we are grateful to the Referee #1 for very useful suggestions. We have incorporated most of their suggestions and the responses are in italics following each of the questioning or suggestion.*

Review report of the manuscript entitled "Diurnal mesospheric tidal winds observed simultaneously by meteor radar in Costa Rica (10°N, 86°W) and Cariri (7°S, 37°W)" by R. A. Buriti et al. The paper presents comparative features of mean winds and tides in the MLT between two low latitude sites located in the opposite hemispheres using meteor radar winds. They further compared the observations with GSWM model. The topic of the present paper is interesting to the scientific community. However, the present manuscript contains several serious issues pertained to technical and scientific aspects. In view of presentation and language/vocabulary it can be considered as a draft and requires substantial modification to improve it to a communicatory level. Anyways, I am detailing my comments/suggestions below.

**Major points**
The theme of the paper is not clear as the title/abstract/conclusion points out the diurnal tide characteristics, but the results start with SAO and AO of the mean winds. Authors should decide the theme and organize the manuscript accordingly.

*Yes, we agree. The main idea of this work is to compare diurnal tides observed by meteor radars from two different places, Costa Rica (10°N) and Cariri (7°S). They are 5800 km apart. We decided to present the background wind because we believe it was important information in case of reader wishes to compare the winds in CR and CA. Also, the reader can use that information to compare to CIRA and/or HWM models. This is because we plotted the monthly average.*

In the present comparative study the observational interval is too short (9.5 months) to present seasonal behaviour over a year which is another weak point of the paper. Derived AO features are questionable, especially at CR due to short data length. Since CA and GSWM database is longer, results can be shown for the missing months of the year.

*You are right. We presented the annual oscillation results but with some restrictions. In the text we discussed about it. We changed the text to remove any information about AO in the manuscript. We have maintained the figure of the background wind and, briefly, we described the semiannual oscillation even with 10 months of observation.*

Fig. 2: It seems that the authors carry out least square fit considering both SAO and AO simultaneously in the fitting function (as found in the caption of the figure). It is not clear how the authors decipher the individual SAO/AO amplitude/phase from the figure.

*We decided that the importance of background wind should be secondary. So, we just presented and discussed the figure of the background wind and semiannual oscillation results briefly. Actually, in this second analysis, we fitted only 182.5 days to the original data according to each altitude gate.*

Fig. 7: Here GSWM data are shown over whole latitude range of ±12°, which is not necessary. Also, authors limit to only two height bins. Instead of the present figure the authors can show four subplots using contours, estimating the difference between radar and GSWM amplitudes incorporating total
MLT range of zonal/meridional for CR/CA. Deviation of amplitude can provide better clarity
regarding the scientific point authors attempt to express.

*The idea to present GSWM from 12°S to 12°S was to emphasize the minimum amplitude of zonal and meridional components close to the equator. We changed to contour plots. Now we present the percent variance from April to January and from 82 to 98 km height.*

Language needs significant improvement to bring coherence in the results/interpretations. It hinders spread of the essence of the work to the readers. Vocabulary, tense and preposition should be corrected. It will be a good idea to check with a native English speaker.

**Other points**

L. 21: "In regard to phases, agreement between meridional tidal phases at the two sites was excellent". The statement is incorrect. The meridional tide phases of two sites are almost opposite (~ 12 h difference).

*We changed to "Considering phases calculated by GSWM-09 between 82 and 98 km, the difference between CR and CA should be, on average, 8.7 ± 0.6 hour. The GSWM-00, an earlier version of the model that does not include non-migrating tides, shows the difference between CA and CR should be 12.0 ± 1.6 hr, on average. That result is close to the observations."*

Fig. 1: Assign marks, i.e., a, b, c, d. *Done*

Fig. 2: Wind data should be shown along with the fit. *That figure was removed.*

L. 96-116: The amplitudes of SAO and AO contain temporal variability. Authors' statement of specific amplitude at a particular altitude raises confusion as it does not make any sense. Same applies to phases.

*It was removed from the text.*

L.25: correct "heat latent release" to "latent heat release" *Done*

L. 37: Replace "diagnostics" by "parameters" *Done*

L. 42: Correct "has shown…." to "showed…." *Done*

L. 44: Correct "have shown that…." to "showed that…." *Done*

L. 51: Delete "atmospheric" from the statement "mesospheric atmospheric dynamics…" *Done*

L. 86: Correct "variationson scalesof months…" to "variation on the time scale of month…" *Done*

L. 96: What do the authors mean by the term "A long-term yearly harmonic analysis"?

*It was changed to "A semiannual harmonic analysis"*

Amplitude of SAO and AO are found to be very small ~ 1-2 m/s. Information related to the uncertainty of the radar winds should be discussed in section 2.

*We wrote some lines about it. I hope it can clarify the uncertainty of the radar.*

*"Concerning the standard deviation of amplitude and phase, it is important to note that, each hour of composite day, includes several thousands of meteor trails detected by the radar. The consequence of this is that the errors in amplitude and phase can be estimated in less than 10% and 1 hour, respectively."*

Do the values shown after " ± " represent standard deviations? *Yes, it represents the standard deviation.*

L.116: "On average, the phase is close to 6 doy (January 6th)". The meaning is not clear as the range in Fig. 2 is shown within 90-390 doy.

*We have removed all text concerning to annual oscillation.*

L. 136: correct "maximum intensity to the south…" to "maximum magnitude towards south…". *Done.*

L. 140: correct "do not coincident…" to "do not coincide…" *Done*

L. 183: "upward propagation…". Please mention upward propagation of what.

*We changed to "upward propagation of diurnal tide is clear, especially in CR where the phase decreases as the altitude increases."*

L. 204: Correct "presented previously" to "mentioned ealier". *Done*

L. 206: Correct "thing that was…" to "feature…" *Done*

L. 210: Delete the statement "So the phases are close to 12 hours different." *Done*

**Discussion:** The first paragraph provides information related to the GSWM and it should be shifted to section 2 with some modifications.

L. 301: "Some reports have proposed that latent heat release is important to semidiurnal tides." Please provide reference. *Done*

L. 307: "So, convective activity could explain the difference of zonal component behavior between CR and CA". Since the tide is more prominent in the meridional wind such influence of convective activity should also be visible in that. Authors' claim of convective effect only on zonal component is not acceptable. Similar statement is also mentioned at the end of the abstract section.

*Yes, we changed the text including meridional component because it also presented a large variation that was not observed in CA and was not predicted by the model.*

The abbreviations CA and Cariri are used interchangeably throughout the manuscript. Please adhere to either CA or Cariri. Same applies to Costa Rica. *Done.*

Angeo-2019-134

**Response (*in italic*) to Referee #2.**
The paper presents some new radar wind measurements in equatorial mesosphere and discussion on the diurnal tidal modulations in zonal and meridional winds. Al- though the description is relatively clean and easy to follow, my major concern is the comparisons with GSWM. The model version utilized in the paper is GSWM00 that does not include non-migrating tide. However, in equatorial region, the some non- migrating diurnal tides, such as DE3 and DE2, are equally important compared with diurnal tidal component, as the author mentions in the paper. Some of the major discrepancies in the paper are most likely due to this issue. There are more complete GSWM versions available, such as GSWM02, GSWM09 etc. that have non-migrating tidal components included. I strongly suggest the author to include the latest model predictions in the next version of the paper, in addition to the current GSWM00 comparison, and expand the discussion based on the new comparison results. Minor comments: 1. line 24-26. I do not think the local weather can affect the tidal feature that much in mesosphere at the same location, since the tidal waves are global scale waves, propagating horizontally with very fast phase speed. The local observations in upper atmosphere reflect the tidal forcing several thousand km away, so the connection to the local tropospheric weather is not straight forward. 2. Line 43. I would add some more references on the tidal comparisons work between ground-based measurements and model. Here are a couple of them: Ward et al., 2010 and Yuan et al., 2006. 3. Line 111. "very low value" sounds strange, may be replaced with "less value". 4. Line 127. Maybe consider to add some brief discussion, from theoretical point of view, why there is such big difference between southern and northern hemisphere equatorial region. 5. Line 196-197, please consider to remove "We also have ... not included in this work" 6. Line 254, see my comment above. Unless there is some solid reference on this topic, I would hesitate to make such statement.

*First of all, we are grateful to the Referee #2 for very useful suggestions. We have incorporated most of their suggestions and the responses are in italic font.*

*The suggestion to use GSWM-09, the last version of the GSWM, is very good and welcome because the manuscript does a reference to non-migrating tide. Initially, we worked with version 00 because its results were available for us. The last version is not available on the internet. We thank Maura Hagan and Xiaoli Zhang for sending the result of the model to make this work. An interesting feature observed was concerning the two versions of the model: GSWM-00 is closer to the observed meridional diurnal phase difference between CR and CA than GSWM-09. We only commented on this result in the text. The idea was not comparing models.*

1. line 24-26. I do not think the local weather can affect the tidal feature that much in mesosphere at the same location, since the tidal waves are global scale waves, propagating horizontally with very fast phase speed. The local observations in upper atmosphere reflect the tidal forcing several thousand km away, so the

connection to the local tropospheric weather is not straight forward.

*It makes sense that the response of tides in the mesosphere does not come from local where we have observed wind. Actually the climate in a specific region is a consequence of a series of phenomena that took place in another part of the world. We decided to explore a little bit the response of ENSO to Costa Rica and Cariri.*

2. Line 43. I would add some more references on the tidal comparisons work between ground-based measurements and model. Here are a couple of them: Ward et al., 2010 and Yuan et al., 2006.

*You are completely right. The work of Ward et al., 201, presented comparisons between model and observations from a series of instruments installed in many places in the world and included, also, satellite data. Both references were cited in the work.*

3. Line 111. "very low value" sounds strange, may be replaced with "less value".

*Yes, sounds strange. We changed.*

4. Line 127. Maybe consider to add some brief discussion, from theoretical point of view, why there is such big difference between southern and northern hemisphere equatorial region.

*In the Introduction, we included some lines in order to explain, in general way, why the atmosphere is different according to the hemisphere. We included in the work:*

*"The classical theory of tides is moderately well-established because it neglect, for example, mechanical forcing and dissipation, considering the atmosphere horizontally stratified and isothermal. But many issues about interaction, excitation and temporal variability require further understanding. Those two mechanisms which drive migrating and non-migrating tides, mentioned above, are, basically, dependent on how the solar radiation heats the planet, according to seasonality and distribution of ocean and continental plates on Earth surface, which makes the global heating different to both hemispheres."*

5. Line 196-197, please consider to remove "We also have ... not included in this work" *Done*

6. Line 254, see my comment above. Unless there is some solid reference on this topic, I would hesitate to make such statement.

*You right. That sentence was deleted.*

[revised manuscript text omitted]

---

## Referee Report (RR1)

2nd review of "Diurnal mesospheric tidal winds observed simultaneously by meteor radar in Costa Rica and Cariri" by Buriti et al.

I appreciate the author's effort to address my questions and comments in my first review. The paper is greatly improved and scientifically sound. It would make the paper more insightful if the author could utilize GSWM09 to discuss a little more about the different diurnal tidal wind behaviors at these two sites. For example, the author could generate contour plots of tidal wind amplitude seasonal variations for the migrating tide DW1 and nonmigrating tide DE2 and DE3 at these two locations. This may help identifying the driving force of the distinct tidal wind seasonal changes at CR and CA, making the discussion more completed. The model should have the outputs for these diurnal tidal components.

Some minor comments:

Line 42, Not sure what "These two mechanisms" are. Please specify.

Line 53-55, please provide the reference on this statement.

Line 93, remove "this software" and replaced it with "it".

Line 101, please consider to replace "not exactly coincident with" with "closest to" .

Line 102, delete "but they close".

Line 105, delete "will here".

Line 107, "Data in February and March are missing for CR" due to…? Please specify.

Line 146, replace "between the" with "for both".

Line 168, please consider to replace "height was quite higher" with "was considerably larger than the model outputs".

Line 169, replace "Month" with "The results".

Line 170, replace "seems to approach to" with "generally agree with".

Line 171-172, please consider to delete "also presented" and modify the sentence as "showed a large blue area in November-January which indicates….".

Line 175-176, delete "of the two,".

Line 179, add "some" in front of "interesting results".

Line 182, add "with long vertical wavelength" after "non-migrating tide".

Line 189-190, "show evidence of model…". I am not sure I fully understand this statement. Please consider to rephrase.

Line 192, about the difference between the model and experimental results, considering the uncertainties in both the measurement and the model, these are not big differences and not unacceptable.

Line 215, replace "observational" with "radar".

Line 240, Should be "The geographic conditions".

Line 243, delete "could effects".

Line 245, delete "to be".

---

## Author Response (AR2)

[revised manuscript text omitted]

Response to the Reviewer #1 – second review.

*First of all, thank you very much again. Your suggestions surely have improved the manuscript.*
*Referee's comments are in standard font, our responses are in itallics.*

2nd review of "Diurnal mesospheric tidal winds observed simultaneously by meteor radar in Costa Rica and Cariri" by Buriti et al.

I appreciate the author's effort to address my questions and comments in my first review. The paper is greatly improved and scientifically sound. It would make the paper more insightful if the author could utilize GSWM09 to discuss a little more about the different diurnal tidal wind behaviors at these two sites. For example, the author could generate contour plots of tidal wind amplitude seasonal variations for the migrating tide DW1 and nonmigrating tide DE2 and DE3 at these two locations. This may help identifying the driving force of the distinct tidal wind seasonal changes at CR and CA, making the discussion more completed. The model should have the outputs for these diurnal tidal components.

*You have raised an interesting point. Initially we worked with the GSWM-00 model. Then, according to a reviewer suggestion, we changed to the GSWM-09 because it considers nonmigrating oscillation modes too. So, we contacted Dr. Xiaoli from University of Colorado in order to get the amplitude and phase of diurnal and semidiurnal components, which including migrating and nonmigrating modes, generated by the model. Because we did not run the model, it is impossible for us generate contour plots according to your suggestion. Your idea is valid and very interesting, but it is not applicable for now. Maybe, we can work on it in the future. The idea of this manuscript was only to present the results observed at both location and present some modest discussion about it.*

Some minor comments:
Line 42, Not sure what "These two mechanisms" are. Please specify.

*Resp: The two mechanisms are forcing and dissipation. We have changed the text to suit.*

Line 53-55, please provide the reference on this statement.

*Resp. We have provided three references: two of them were already cited in the text, and a new one by Deepar et al., 2006 has been added. All show irregularity of phase with altitude.*

Line 93, remove "this software" and replaced it with "it".

*Resp: Done.*

Line 101, please consider to replace "not exactly coincident with" with "closest to" .

*Resp: Done.*

Line 102, delete "but they close".

*Resp: Done.*

Line 105, delete "will here".

*Resp: Done.*

Line 107, "Data in February and March are missing for CR" due to…? Please specify.

*Resp: we added "because the meteor radar presented technical problems".*

Line 146, replace "between the" with "for both".

*Resp: Done.*

Line 168, please consider to replace "height was quite higher" with "was considerably larger than the model outputs".
*Resp: Done.*

Line 169, replace "Month" with "The results".
*Resp: Done.*

Line 170, replace "seems to approach to" with "generally agree with".
*Resp: Done.*

Line 171-172, please consider to delete "also presented" and modify the sentence as "showed a large blue area in November-January which indicates….".
*Resp: Done.*

Line 175-176, delete "of the two,".
*Resp: Done.*

Line 179, add "some" in front of "interesting results".
*Resp: Done.*

Line 182, add "with long vertical wavelength" after "non-migrating tide".
*Resp: Done.*

Line 189-190, "show evidence of model…". I am not sure I fully understand this statement. Please consider to rephrase.
*Resp: Yes, you are right. So, we changed the sentence to "Because the zonal diurnal phase in CR and CA, normally, showed undefined behavior with height, only 4 months were available to determine the vertical wavelength using the above criteria."*

Line 192, about the difference between the model and experimental results, considering the uncertainties in both the measurement and the model, these are not big differences and not unacceptable.

*Resp: We decided to remove the last sentence of this paragraph ("Nevertheless, it seems that the GSWM-09 does overestimate the vertical wavelengths")*

Line 215, replace "observational" with "radar".
*Resp: Done.*

Line 240, Should be "The geographic conditions".
*Resp:We changed to "The geographic and climate conditions"*

Line 243, delete "could effects".
*Resp: Done.*

Line 245, delete "to be".
*Resp: Done.*

Response to the Reviewer #2 – second review.

*First of all, thank you very much again. Your suggestions surely have improved the manuscript. Our answer and comments are in italic font.*

Review report of the revised manuscript entitled "Diurnal mesospheric tidal winds observed simultaneously by meteor radar in Costa Rica (10° N, 86° W) and Cariri (7° S, 37° W)" by Buriti et al. Although the authors revised the manuscript, it still requires further revision as there exist a couple of issues that weaken the content of the paper.

Major points
Abstract should be modified with essential information in concert with the present findings. Similarly, the section 5 (conclusions) should be more precise. In my view section 5 is rather a summary of the present work.

*Answer: Thank you, we have rewritten those parts.*

L.114-121: The description of the semiannual oscilation parameters, e.g., amplitude phase etc. directly from the wind (Figure 1) looks incongruous. Authors should restructure this part in order to express the information clearly and maintain the flow of the text.

*Answer: Yes, we agree that there is, a priori, incongruence between text and figures. Because we presented only semiannual oscillation, there are two peaks during the year. One is in ~160 doy and other in 342 doy (December 8$^{th}$). It is clear that in December the wind is stronger in CR (in CA the zonal wind is stronger in June). It was showed in the paper of Buriti et. al., 2008, that there is a contribution of annual component to zonal and meridional winds which was not showed here because the data set of CR is less than one year. In the first version of the manuscript we presented both components (annual and semiannual) but we called special attention to the annual harmonic results. So, in the new version we only presented semiannual results from a simple harmonic analyses considering only semiannual component. We mentioned December in the text too. Hopefully this makes the description clear.*

Fig. 4: Large percentage deviations between the model and observations persisting over significant time span implies poor comparison among them. Authors should address this issue carefully.

*Answer: In the first version of the manuscript we chose some specific altitudes and months to compare observation and model. It was suggested that we could make a plot where we could compare all data together. This kind of comparison, considering time and altitude, can make the difference increase considerably. Sometimes, we guess, a general comparison is more adequate to this kind of work. But, important points that we have discussed in the text are clearly showed in Fig. 4.*

Other points
L. 70: Correct "5 receiver antenna…" to "5 receiver antennas…."
*Done*

L. 82: The error in amplitude should be mentioned in absolute value rather than percentage.

*Answer: The software we have used to analyses meteor wind observed by meteor radars gives us the error as a percentage. In term of absolute values, we could say that the error, varies from 1 to 4 m/s. This error is associated, basically, with the number of days used to make the composite day of each month.*

L. 93: Remove "software" from the sentence.
*Done*

L. 114: Correct "A semiannual harmonic analysis" to "A harmonic analysis to derive semiannual oscillation…"
*Done*

L. 116: Correct "doy (day of year) 160" to "160 doy (day of year)".
*Done*

L. 143: Correct "compared to CA" to "compared to CR".
*Done*

L. 161: Correct "will be show" to "will be shown".
*Done*

L. 165: The caption of the figure 4 indicates percentage deviation. However, the text mentions "percent variance".
*Done*

L. 167: Correct "model is bigger" to "model is higher".
*Done*

L. 208: The unit of the vertical wavelength should be "km".
*Done*

L. 226: The authors cursorily mentioned "good agreement" between the model and observations and hence the information is not clear.

*Answer: We changed this sentence to " They show in Figure 6 of their work a composite-year monthly-mean zonal and meridional diurnal amplitudes as a function of month and altitude which present a good agreement to the GSWM-09 model "*

*We also changed part of the paragraph in order to make it clearer to the reader.*

L. 235: "That regular variation was not observed in zonal diurnal phase, in contrast to the meridional one". It seems that the authors mean radar observations here.

*Answer: This has been changed to "That regular variation was not observed by the radar in zonal diurnal phase, in contrast to the meridional one."* L. 243: There is a repetition in the sentence "could effects could affect…"
*Answer: We have corrected it.*

L. 260: The relationship between the convective activity and rainfall causing higher amplitude of tide in January than July remains obscure and should be elaborated further.

*It is well established that nonmigrating tides depend on how Sun heats the planet, which depends in turn on seasonality and geographic conditions. Thermal forcing (water vapor heating and latent heat) is associated, in general, with convection in the troposphere. The model, considers normal atmospheric conditions according to month and location. Plots of precipitation anomaly from NOAA show that a positive anomaly happened in December in CR. So, if precipitation is associated with deep convection, we could suggest that the increase of zonal amplitude in December in CR was because tropospheric latent heat and water vapor absorption were relevant in that period. So, we need to be cautious and wish to suggest this, but we don´t want to affirm too definitely that it is true. Probably we could go deeper in this subject if we investigate all those parameters simultaneously during a couple of years. But we do not wish this to be too much of a focus for this paper. So, the idea of this work was, in advance, to present interesting results from meteor radars from two different places with roughly complementary latitudes but for the same time period of observation.*

I am sorry to reiterate that the English is not up to the mark and should undergo thorough checking. The language requires substantial refinement to reach a satisfactory level. Grammar needs to be corrected at various places in the text carefully.

*Answer: We tried to improve the level of the text.*